# A Novel P300 Classification Algorithm Based on a Principal Component Analysis-Convolutional Neural Network

**Feng Li [1,2,†], Xiaoyu Li [1,2,†], Fei Wang [1,2,3,\*] , Dengyong Zhang [1,2] , Yi Xia [1,2] and Fan He [1,2]**

[1]  School of Computer and Communication Engineering, Changsha University of Science and Technology, Changsha 410114, China; Lif@csust.edu.cn (F.L.); csustecolee@foxmail.com (X.L.); zhdy@csust.edu.cn (D.Z.); csustxy@gmail.com (Y.X.); hf0208@stu.csust.edu.cn (F.H.)

[2]  Hunan Provincial Key Laboratory of Intelligent Processing of Big Data on Transportation, Changsha University of Science and Technology, Changsha 410114, China

[3]  School of Software, South China Normal University, Guangzhou 510631, China

\*  Correspondence: scutauwf@foxmail.com

†  These authors contributed equally to this work.

**Abstract:** Aiming at enhancing the classification accuracy of P300 Electroencephalogram signals in a non-invasive brain–computer interface system, a novel P300 electroencephalogram signals classification algorithm is proposed which is based on improved convolutional neural network. In the data preprocessing part, the proposed P300 classification algorithm used the Principal Component Analysis algorithm to not only remove the noise and artifacts in the data, but also increase the data processing speed. Furthermore, the proposed P300 classification algorithm employed the parallel convolution method to improve the traditional convolutional neural network framework, which can increase the network depth and improve the network's ability to classify P300 electroencephalogram signals. The proposed algorithm was evaluated by two datasets (the dataset from the competition and the dataset from the laboratory). The results show that, in the dataset I, the proposed P300 classification algorithm could obtain accuracy rates higher than 95%, and achieve one of the best performances in four classification algorithms, while, in the dataset II, the proposed P300 classification algorithm can get accuracy rates higher than 90%, and is superior to the other three algorithms in all ten subjects. These demonstrated the effectiveness of the proposed algorithm. The proposed classification algorithm can be applied in the actual brain–computer interface systems to help people with disability in the daily lives.

**Keywords:** brain–computer interface (BCI); electroencephalogram (EEG); P300

## 1. Introduction

Brain–computer interfaces (BCI) can provide a direct communication method between the brain and a computer or other external devices [1–3]. There are several types of electroencephalograms (EEG) signals used in BCI, such as P300 potential [4], steady state visual evoked potential (SSVEP) [5], motor imagery (MI) [6], and so on. Specifically, P300-based BCI is one of the most common BCI systems, as the P300 potential is easy to be stimulated. Compared with other signals, the P300-based BCI system has some advantages: (1) P300 signal is extremely easy to measure and non-invasive; (2) less training time; (3) suitable for most subjects, including those with severe neurological diseases; and (4) users only need to provide a simple control signal [7]. It can implement a variety of different functions, and can even be used in the home of people with disability [7,8].

Farwell and Donchin of the United States introduced the first P300-based character input system in 1988 [9,10], which has been applied until now. The system contained a 6*6 matrix of visual stimulation interface, which was composed of English letters, numbers, and spaces. Before the experiment, the subjects were told that a specified character in the visual stimulator was the target character, and each experiment randomly assigned a character. During the experiment, the subjects were asked to keep an eye on the target character position in the visual stimulator, while any row or column in the visual stimulator flashed randomly. When the target character's row or column was flashing, a positive potential (called P300 ERP) related to the event could be detected in the subject's scalp (about 300 ms after receiving the stimulus); if not, the detected EEG data were non-P300 event-related potentials (N-P300 ERP) [11]. In addition to this standard speller system, there are other paradigms, such as row-column (RC) paradigm [12], single character (SC) paradigm [13], region-based(RB) paradigm [14], and so on. For all these systems, how to identify quickly and accurately is critical to improving the performance of BCI systems.

Due to the collected P300 signals often being high dimensional and feature dependent, some methods were proposed to enhance the feature extraction. PCA (Principal Component Analysis), a principal component analysis method, is widely used in feature extraction and data dimensionality reduction. The principle of PCA is to transform the original signal matrix into a covariance matrix [15] through linear transformation [16], and obtain a new signal matrix by filtering the eigenvalues and eigenvectors of the matrix [17]. The new signal matrix retains some of the most important original signal features in the original signals matrix, and eliminates noise and unimportant features to achieve the purpose of dimensionality reduction. In recent years, many researchers have applied the PCA to reduce the dimensionality of the obtained EEG signals. Salma Tayeb used different dimensionality reduction algorithms to process the EEG signals for the dimensionality reduction part of the data, and found that the use of PCA for dimensionality reduction of P300 signals performed best, compared to independent component analysis (ICA) and linear discriminant analysis (LDA) [18]. Kundu and Sourav used PCA to reduce the dimensionality of P300 signals, and then used SVM to classify the reduced-dimensional signals. PCA reduced the computational burden of weighted classifiers and speeds up the classification speed [19]. Like combined multi-scale filters and PCA to classify EEG signals, the classification accuracy can reach 91.13% [20]. PCA fits in a brain–computer interface, especially the P300 brain–computer interface.

Previous works on P300 classification mainly employed traditional machine learning algorithms, such as support vector machine (SVM), linear discriminant analysis (LDA), and so on. Rakotomamonjy divided P300 EEG signals into several equal parts, and then used SVM to train corresponding classifiers for each part, which improved the accuracy of P300 EEG signals recognition [21]. Chandra S. Throckmorton proposed a Bayesian P300 recognition method to complete classification by determining the maximum regression target probability value. Although the accuracy of classification was improved, it took too much calculation time [22]. With the rapid development of deep learning, many scholars began to use convolutional neural network (CNN) to classify P300 EEG signals [23]. Cecotti realized the use of convolutional neural networks in deep learning to recognize and classify P300 EEG signals. He used convolutional layers to separate the time and space domains of P300 EEG signals. The convolutional neural network is fast but very easy to overfit, which affects the accuracy of recognition [24]. Lawhern Vernon used compact convolutional neural network (CNN) to classify four types of EEG data including P300 EEG signals, and the results shown that the compact CNN has the best classification effect on P300 EEG signals [25]. Sobhani proposed to use deep belief network (DBN) to classify P300 EEG signals which were extracted from each channel, but only a few subjects have good recognition accuracy [26]. Maddula proposed a 3D recurrent convolution neural network (3DRCNN) based on a recurrent convolution neural network (RCNN) to classify P300 EEG signals. It was processed into 3D-EEG signals, and then input them into the RCNN to achieve nice classification [27]. LIU improved the convolutional neural network on the basis of Cecotti's algorithm which was named BN3 algorithm, taking the batch normalization (BN) layer and the dropout (DP)

layer to deepen the network layers and overcome the problem of overfitting. The BN3 algorithm achieved good results in classification, but still need to improve the recognition accuracy when the number of experiments are reduced [28].

In this paper, we combined PCA algorithms with new convolution neural network framework to complete the P300 signals classification and recognition (named PCA-CNN). Specifically, the PCA algorithm was used to reduce the dimension of the EEG signal, which not only reduced the calculation time, but also improved the signal-to-noise ratio of the data. The new convolution neural network framework improved single convolution kernel model of traditional convolutional neural network. It contacted multiple convolution kernels to classify P300 EEG signals in the convolution layer, which improved the recognition ability of the convolutional neural network. Two datasets (competition data and self-collected data) were used to verify the effectiveness of the algorithm. The results indicated that the proposed algorithm had a significant effect on the recognition accuracy of P300 EEG signals.

## 2. Method

### 2.1. The Dataset

Two sets of experimental data were analyzed in this paper. One was the dataset in the BCI Competition III provided by the Wadsworth Research Center NYS Department of Health [29]. The other was provided by South China University of Technology using a different paradigm. There are two and ten subjects, respectively, in the two datasets. All subjects are healthy persons, who were selected randomly. Specifically, the two subjects (A and B) were chosen from five people in the public BCI Competition III in 2004. The other ten subjects in the dataset II were recruited randomly and participated in the brain–computer interface experiments for the first time, provided by the South China University of Technology.

Dataset I: The graphical user interface (GUI) of the competition was presented in Figure 1a, which is a 6*6 character matrix. When the experiment began, each of the 12 rows and columns flashed randomly. A flashing lasts 100 ms and the interval between two flashing is 75 ms. A subject was asked to focus on the target character, and silently count the flashing repetitions of the row and column containing the target character. Each row or column repeats 15 times when outputting one character. The dataset was consisted of one training (85 characters) and one test (100 characters) sets for each of the two subjects A and B. All EEG signals were collected by a 64-electrode scalp, which were bandpass filtered from 0.1–60 Hz and digitized at 240 Hz. The information details can be found in the BCI competition webpage.

Dataset II: The second dataset was collected in the laboratory from South China University of Technology using a 4*10 paradigm (see Figure 1b). Different from the first dataset, each character flashed separately and randomly. A flashing lasts 100 ms and the interval is 30 ms. A subject was asked to focus on the target character, and silently count the flashing repetitions of the target character. Each character in paradigm repeats 10 times when outputting one character. The dataset was consisted of one training (20 characters) and one test (30 characters) set for each of ten subjects. All EEG signals were collected by a 32-electrode scalp, which were bandpass filtered from 0.1–60 Hz and digitized at 250 Hz.

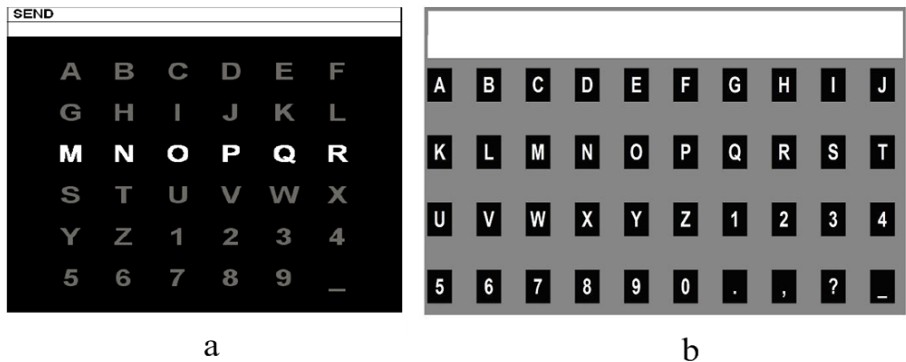

**Figure 1.** Experimental set-up. (**a**) 6*6 GUI used in the brain–computer interfaces (BCI) competition; (**b**) 4*10 graphical user interface (GUI) used in the laboratory.

## 2.2. Data Preprocessing

As the raw EEG signals are weak and mixed with non-EEG signals or background noise, the raw data should preprocess first. In order to remove the influence of these interference information [30], we used the 8th-order bandpass Butterworth filter [31] to filter the collected raw data and intercept the potential frequency to 0.1–20 Hz. Then, the number of positive samples should be increased before the next step, in order to prevent the classification problems caused by the imbalance of positive and negative samples. We will increase the number of P300 EEG signals to the number of non-P300 EEG signals; that is, copying P300 EEG signals so that the number of P300 EEG signals is the same as or close to non-P300 signals.

## 2.3. PCA Algorithm

PCA transforms the original data into a set of linearly independent data vectors in various dimensions through linear transformation, which can be used to extract the main feature components of the data and often be used for dimensionality reduction of high-dimensional data. Therefore, before EEG signals were input into the neural network, we used the PCA algorithm to reduce the signal dimension. After preprocessing of the raw signals, a data matrix X is obtained, of which the abscissa and ordinate are the time and space domains, respectively. We input the matrix X into the PCA algorithm and calculated the covariance matrix (Cov(X)); then, the eigenvalue eigenvectors of the covariance matrix were obtained. We could select a matrix of eigenvectors corresponding to the features with the largest eigenvalues. In this way, the data matrix could be transformed into a new space, and the dimension reduction of the data features could be realized. Through dimensionality reduction using PCA, the size of feature vectors changed from 64*240 and 30*160 to 64*120 and 30*80 (two datasets respectively). The mathematical formula is as follows:

Input the matrix $X(x1, x2, x3, ..., xn)$, and reduce the matrix X to K dimensions, $0 \leq K \leq n$.

Find the covariance matrix of matrix X:

$$\text{Cov X} = \frac{1}{n}XX^T \tag{1}$$

Find the eigenvalues and eigenvectors of the covariance matrix CovX:

$$\text{Cov X} = \Lambda L \tag{2}$$

where $\Lambda = diag[\lambda 1, \lambda 2, \ldots, \lambda n]$ is the eigenvalue of the *X* covariance matrix, and *L* is the eigenvector of the *X* covariance matrix.

Sort the eigenvalues $\Lambda(set\lambda1 \geq \lambda2 \geq ... \geq \lambda n \geq 0)$ from large to small, and select the largest k of them. Then, use the corresponding k feature vectors as row vectors to form a feature vector matrix $P$ and the data are transformed into a new matrix $Y$ constructed by k feature vectors:

$$Y = PX \tag{3}$$

where $Y$ is the matrix after dimensionality reduction.

### 2.4. Parallel Convolutional Network

In this paper, we proposed an improved neural network architecture (as shown in Figure 2). The specific network contained 9 layers, the parameters of which were illustrated in Tables 1 and 2. L1 is the data input layer. L2 is the spatial domain convolution layer. L3 to L6 are the parallel convolution layer [32,33] to extract time domain features. L7 is the pooling layer. L8 and L9 are fully connected layer and softmax layer. The computations in each layer (following, as an example, Table 1 Parameters setting) were described in detail as follows:

L1: The input layer loads the pre-processed EEG data into the network and uses $a$ to represent the data tensor transmitted to the neural network.

L2: Convolution layer, which is a spatial filter for all channels of the input signal, can improve the signal-to-noise ratio [34], and remove redundant signals in the spatial domain. The formulas are as follows:

$$a_i^1(j) = \sum_{i=1}^{i=64} a_i^0(j)w_0 + b_0 \tag{4}$$

where $a_i^1(j)$ is output data for L2, $i$ ($1 \leq i \leq 64$) denotes the Spatial dimension, and $j$ ($1 \leq j \leq 120$) denotes the time dimension. $w_0$ denotes the weight, $b_0$ denotes the deviation (all $w_n$, and $b_n$ $(0, 1, 2, ..., n)$ denotes the deviation of different values below).

L3 and L4: convolutional layer and dropout layer. This layer is arranged in parallel by three convolutional layers of different convolution sizes. Each convolution and size is the same. Different convolution kernels can be extracted to different values for the same input: information, increasing the complexity of features. After filtering in the time domain after L2 layer spatial filtering, we use 16*5*1, 16*10*1 and 16*15*1 convolution kernels for convolution. After convolution, we can get 16*1*24, 16*1*12 and 16*1*8 feature vectors; these feature vectors are combined into 16*1*44 feature vectors. A dropout layer is added after the convolutional layer to prevent overfitting in the case of too many model parameters [35]. The formula is as follows:

$$a_j^2(s) = \sum_{j=1}^{j=120} a_j^1(s)w_1 + b_1 + a_j^1(s)w_2 + b_2 + a_j^1(s)w_3 + b_3 \tag{5}$$

$$r^1 = \text{Bernoulli}\left(p^1\right) \tag{6}$$

$$a_j^3(s) = \sum_{j=1}^{j=44} a_j^2(s)r^1w_4 + b_4 \tag{7}$$

where $a_j^2(s)$ is output data for L3, $s$ ($1 \leq s \leq 20$) denotes the depth of convolution kernels, $j$ denotes the time dimension, and $a_j^3(s)$ is output data after dropout layer. $r$ and $p$ denote the dropout value, $r = 0.5$.

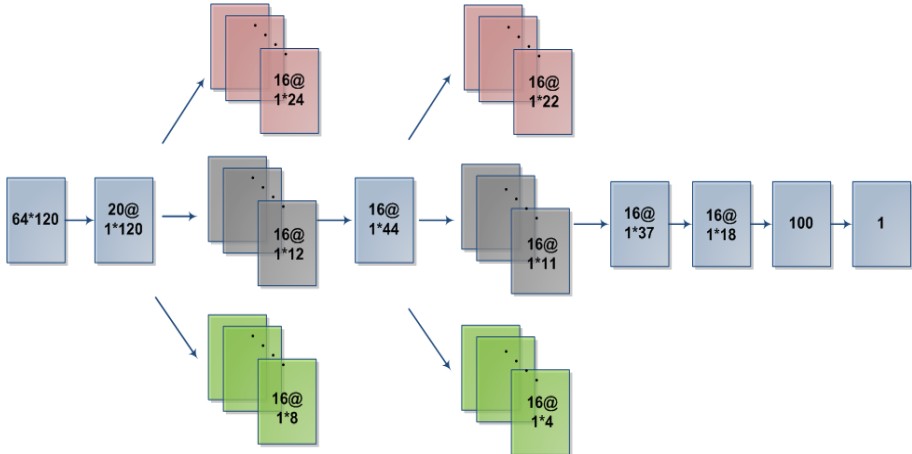

**Figure 2.** The framework model of the 9-Layer convolution neural network used for classification. The feature map is of dimension Depth @ Height * Weight (such as 16@1*24), layers with colors indicate that different size convolution kernels are used. The parameters in this figure are an example of the parameters in Table 1.

L5 and L6: Structures of these two layers are the same as that of L3 and L4. These layers are convolved with 16*2*1, 16*4*1, and 16*11*1 convolution kernels, which can get 16*1*22, 16*1*11, and 16*1*4 feature vectors, and are combined into 16*1*37 feature vectors. A dropout(DP) layer is added after the convolution layer, and the value is 0.5. The calculation formula is as follows:

$$a_j^4(s) = \sum_{j=1}^{j=44} a_j^3(s)w_5 + b_5 + a_j^3(s)w_6 + b_6 + a_j^3(s)w_7 + b_7 \tag{8}$$

$$r^2 = \text{Bernoulli}\left(p^2\right) \tag{9}$$

$$a_j^5(s) = \sum_{j=1}^{j=37} a_j^4(s)r^2 w_8 + b_8 \tag{10}$$

L7: The pooling layer consists of a pooling filter of size 2, which is used to reduce the parameters of the network.

L8 and L9: Fully connected layer and softmax layer. Data after L7 which are connected with 100 neurons, and then classification by softmax layer. We use rectified linear unit (ReLU) as the activation function [36]. $a^6(k)$ and $a^7(k)$ denote the output data after fully connected(FC) layer. $k$ is the number of feature maps. The calculation formula is as follows:

$$a^6(k) = \sum_{k=1}^{k=288} a^5(k)w_9 + b_9 \tag{11}$$

$$a^7(k) = relu\left(\sum_{k=1}^{k=100} a^6(k)w_{10} + b_{10}\right) \tag{12}$$

When the calculated network output probability is greater than or equal to 0.5, the current input signals are determined to be P300 signals, otherwise not. The judgment is as follows:

$$Q = \begin{cases} 1 & (P \geq 0.5) \\ 0 & (P < 0.5) \end{cases} \tag{13}$$

where $Q$ represents the judgment result and $P$ represents the probability value.

The scintillation of 6 rows and 6 columns in the experiment will be repeated 15 times. This is because the position of the target character can not be accurately determined by single experiment scintillation. The probability values corresponding to multiple row scintillation and multiple column scintillation can be accumulated. The target character can be determined by selecting the rows and columns corresponding to the maximum probability values. The formula is as follows:

$$
\begin{cases}
X = \arg_i \max \sum_{K=1}^{K=n} p(K,i) & (1 \le i \le 6) \\
Y = \arg_i \max \sum_{K=1}^{K=n} p(K,i) & (7 \le i \le 12)
\end{cases}
\tag{14}
$$

$X$ denotes the position of column target characters, $Y$ denotes the position of row target characters, $n$ denotes the number of experiments, $P$ denotes the probability value, $K$ denotes the serial number of experiments, and $i$ denotes the row and column numbers.

Dataset II is a random single flicker of 40 characters, from which the location of the maximum probability value can be selected to determine the target character. The formula is as follows:

$$
Z = \arg_i \max \sum_{K=1}^{K=n} p(K,i) \quad (1 \le i \le 40)
\tag{15}
$$

$Z$ denotes the position of the target character, $n$ denotes the number of experiments, $P$ denotes the probability value, $K$ denotes the serial number of experiments, and $i$ denotes the position number.

**Table 1.** The network parameter settings for dataset I.

| Number of Layers | Input | Convolution Kernel and Operation | Output | Activation Function |
|---|---|---|---|---|
| L1 | 1*64*120 | None | 1*64*120 | None |
| L2 | 1*64*120 | 20*64*1 | 20*1*120 | ReLU |
| L3 | 20*1*120<br>20*1*120<br>20*1*120 | 16*1*5<br>16*1*10<br>16*1*15 | 16*1*24<br>16*1*12<br>16*1*8 | ReLU |
| L4 | 16*1*24<br>16*1*12<br>16*1*8 | Concat and DP | 16*1*44 | ReLU |
| L5 | 16*1*44<br>16*1*44<br>16*1*44 | 16*1*2<br>16*1*4<br>16*1*11 | 16*1*22<br>16*1*11<br>16*1*4 | ReLU |
| L6 | 16*1*22<br>16*1*11<br>16*1*4 | Concat and DP | 16*1*37 | ReLU |
| L7 | 16*1*37 | Maxpool | 16*1*18 | ReLU |
| L8 | 16*1*18 | FC | 100*1 | ReLU |
| L9 | 100*1 | Softmax | 2*1 | ReLU |

**Table 2.** The network parameter settings for dataset II.

| Number of Layers | Input | Convolution Kernel and Operation | Output | Activation Function |
|:---:|:---:|:---:|:---:|:---:|
| L1 | 1*30*80 | None | 1*30*80 | None |
| L2 | 1*30*80 | 20*30*1 | 20*1*80 | ReLU |
| L3 | 20*1*80<br>20*1*80<br>20*1*80 | 16*1*5<br>16*1*8<br>16*1*10 | 16*1*18<br>16*1*10<br>16*1*8 | ReLU |
| L4 | 16*1*18<br>16*1*10<br>16*1*8 | Concat and DP | 16*1*36 | ReLU |
| L5 | 16*1*36<br>16*1*36<br>16*1*36 | 16*1*2<br>16*1*4<br>16*1*9 | 16*1*18<br>16*1*9<br>16*1*4 | ReLU |
| L6 | 16*1*18<br>16*1*9<br>16*1*4 | Concat and DP | 16*1*31 | ReLU |
| L7 | 16*1*31 | Maxpool | 16*1*15 | ReLU |
| L8 | 16*1*15 | FC | 100*1 | ReLU |
| L9 | 100*1 | Softmax | 2*1 | ReLU |

### 2.5. Evaluation

To measure the performance of the algorithms, we used two indices, the accuracy rate and the information translate rate (ITR) [24,37] to compare the proposed PCA-CNN with other CNN algorithms in the literature. The accuracy can evaluate the effectiveness of the algorithm. The accuracy rate in the article is the number of correct classified characters classified compared with the number of total actual test characters. The calculation formula is as follows:

$$T_{acc} = P_n / S_n \tag{16}$$

where $T_{acc}$ is the accuracy rate of character recognitions, $P$ is the number of correct detected characters, $n$ is the number of repeats, and $S$ is the number of total characters.

ITR can display the recognition speed of the test characters by the classification algorithm in bits per minute. The formula is as follows:

$$\text{ITR} = \frac{60(P \log_2(P) + (1 - P) \log_2((1 - P)/(N - 1)) + \log_2(N))}{T} \tag{17}$$

where $N$ represents the number of classes, $P$ represents the accuracy rate of character recognitions, and $T$ represents the time taken for character recognitions [24,28].

In dataset I, as each flash lasts for 100 ms followed by a pause of 75 ms ($12 * (75 + 100) = 2100$) and a pause of 2.5 s between each character epoch, $T$ can thus be defined as:

$$T = 2.5 + 2.1n \tag{18}$$

where $n$ is the number of repeats, $1 \leq n \leq 15$.

In dataset II, as each flash lasts for 100 ms followed by a pause of 30 ms ($40 * (30 + 100) = 5200$) and a pause of 1.2 s between each character epoch, so $T$ can be defined as:

$$T = 1.2 + 5.2n \tag{19}$$

where $n$ is the number of repeats, $1 \leq n \leq 10$.

## 3. Experimental Results

In this paper, we used the accuracy rate and ITR to evaluate the P300 signals detection performance on two datasets of different subjects. In dataset I, there are 85 training and 100 test characters for each subject, each of which is repeated 15 times. Tables 3 and 4 present the test accuracy rates of the proposed PCA-CNN and other classification methods in the literature, including CNN algorithms BN3 [28] and CNN-1 [24], and a traditional SVM algorithm [21], on the datasets I of subjects A and B. Bold numbers in the table indicate the best accuracy rate in the $N(1, 2, \ldots, N)$ repeats. For both subjects, the proposed PCA-CNN is one of the best algorithms in 15 repeats, and can obtain the accuracy rate higher than 95%. Furthermore, the PCA-CNN is superior to other three methods from 7 repeats for the subject A in the dataset I, and from 8 repeats for the subject B in the dataset I.

**Table 3.** The accuracy rate of subject A in dataset I.

| Algorithms | The Number of Repeats | | | | | | | | | | | | | | |
|---|---|---|---|---|---|---|---|---|---|---|---|---|---|---|---|
| | 1 | 2 | 3 | 4 | 5 | 6 | 7 | 8 | 9 | 10 | 11 | 12 | 13 | 14 | 15 |
| **PCA-CNN** | **24** | 37 | 46 | 61 | 71 | **75** | **84** | **86** | **90** | 90 | **92** | **94** | **95** | **97** | **98** |
| **BN3** | 22 | **39** | **58** | **67** | **73** | **75** | 79 | 81 | 82 | 86 | 89 | 92 | 94 | 96 | **98** |
| **CNN-1** | 16 | 33 | 47 | 52 | 61 | 65 | 77 | 78 | 86 | **90** | 91 | 91 | 91 | 93 | 97 |
| **SVM** | 16 | 32 | 52 | 60 | 72 | 71 | 82 | 81 | 82 | 83 | 87 | 88 | 94 | 95 | 97 |

**Table 4.** The accuracy rate of subject B in dataset I.

| Algorithms | The Number of Repeats | | | | | | | | | | | | | | |
|---|---|---|---|---|---|---|---|---|---|---|---|---|---|---|---|
| | 1 | 2 | 3 | 4 | 5 | 6 | 7 | 8 | 9 | 10 | 11 | 12 | 13 | 14 | 15 |
| **PCA-CNN** | 30 | 51 | 57 | 69 | 73 | 74 | 83 | **94** | **95** | **96** | **97** | **95** | 96 | **96** | 96 |
| **BN3** | **47** | **59** | **70** | **73** | 76 | **82** | **84** | 91 | 94 | 95 | 95 | **95** | 94 | 94 | 95 |
| **CNN-1** | 35 | 52 | 59 | 68 | **79** | 81 | 82 | 89 | 92 | 91 | 91 | 90 | 91 | 92 | 92 |
| **SVM** | 35 | 53 | 62 | 68 | 75 | 80 | **84** | 86 | 89 | 91 | 92 | 93 | **96** | 95 | **96** |

In dataset II, there are 20 training characters and 30 test characters for each subject, each of which is repeated 10 times. Figure 3 presents the test accuracy rates of the proposed PCA-CNN and other classification methods in the literature, including CNN algorithms BN3 [28] and CNN-1 [24], and the traditional SVM algorithm [21], on datasets II of 10 subjects. The different color line in the figure records the results of all subjects that used different methods on each repeat. For all subjects, the average accuracy rate of PCA-CNN is higher than the other three algorithms in 4th repeats. Furthermore, the average accuracy rate of PCA-CNN is the best algorithm in 10 repeats, and can obtain an average accuracy rate higher than 90%. The experimental results show that the PCA-CNN is superior to others in character recognition.

We record the information translate rates of datasets I and II in different algorithms. As shown in Figures 4 and 5, the ITR value of the PCA-CNN is higher than that of the other three algorithms (BN3 [28], CNN-1 [24], and SVM [21]) in the maximum repeat. In Figure 4, after the 7th repeat, the ITR value of PCA-CNN is higher than the other three algorithms (BN3 [28], CNN-1 [24] and SVM [21]). In Figure 5, after the 4th repeat, the ITR value of PCA-CNN is higher than the other three algorithms (BN3 [28], CNN-1 [24] and SVM [21]). From the overall results of Figures 4 and 5, the PCA-CNN is faster than the CNN-1 [24] and SVM [21]. These indicate that, on the basis of ensuring the characters recognition accuracy rate, the characters' recognition speed of PCA-CNN is still fast, and the PCA-CNN algorithm has application value.

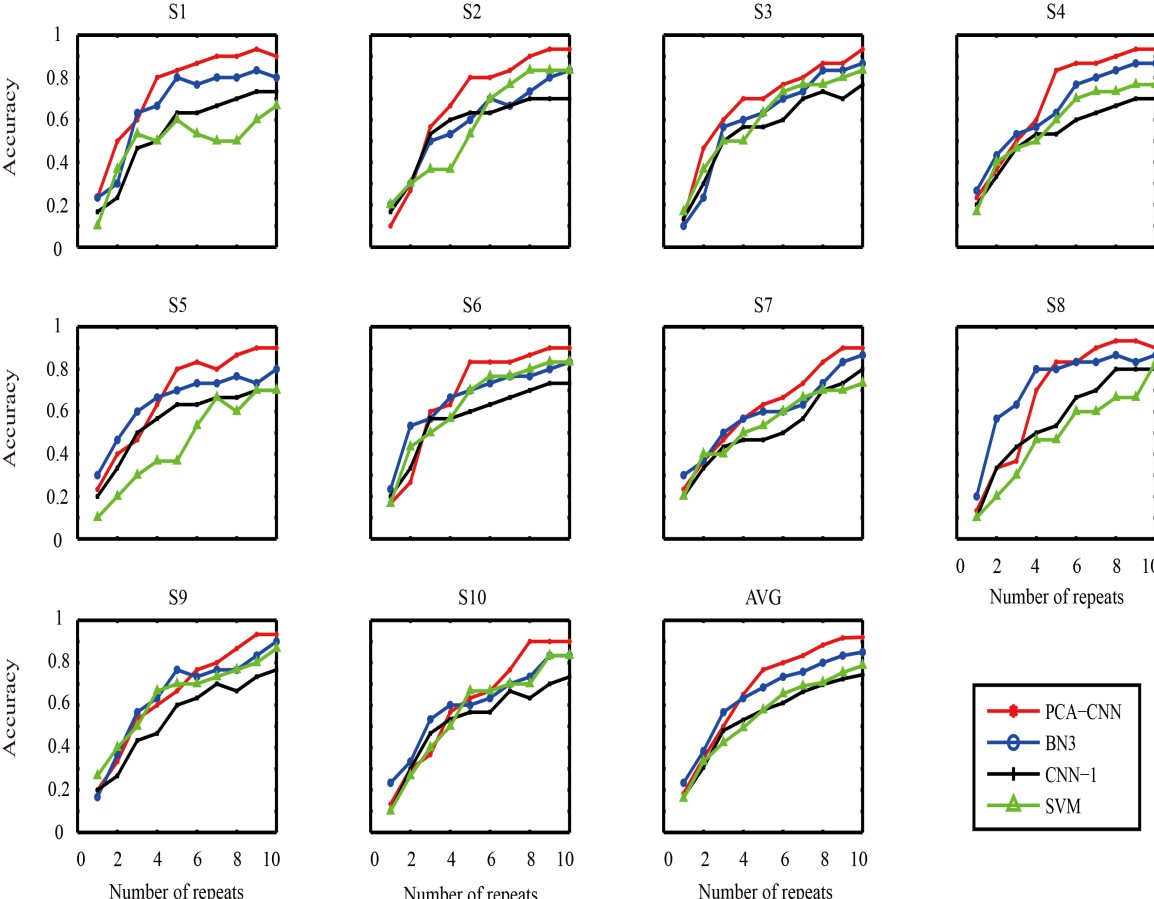

**Figure 3.** The accuracy rates of all ten subjects in dataset II. The vertical axis corresponds to the accuracy rate, and the horizontal axis corresponds to the number of repeats. The color curves represent respectively the different classification accuracy rates of each subject and the average accuracy rate (AVG) of all 10 subjects (S1, S2, ..., S10) tested by four methods.

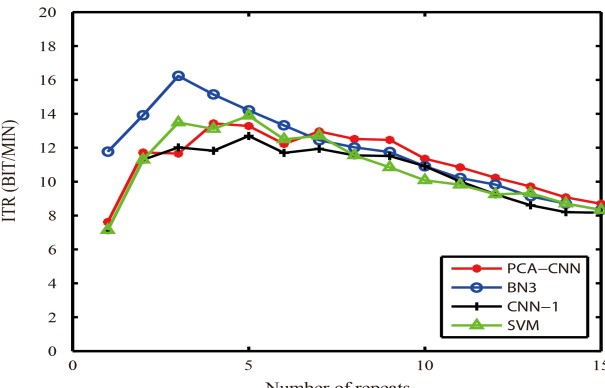

**Figure 4.** The information translate rate graph for four algorithms in the dataset I calculated by the average information translate rates of subject A and subject B.

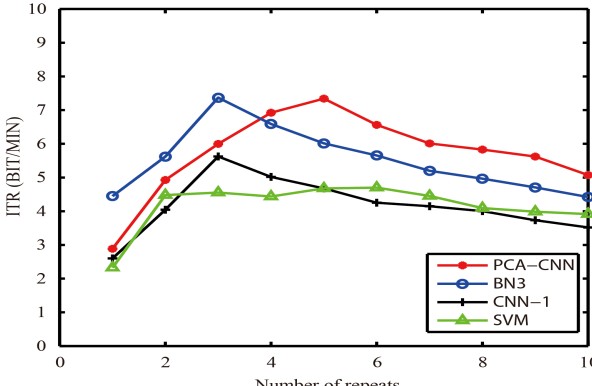

**Figure 5.** The information translate rate graph for four algorithms in dataset II calculated by the average information translate rates of all ten subjects.

## 4. Discussion

This paper proposed a PCA-CNN algorithm to improve the classification performance of P300-based BCI system. The PCA was used to reduce the P300 EEG signals dimension, as PCA could not only removed the noise and unimportant features of P300 EEG signals, but also improved the speed of EEG data processing. Furthermore, we used the improved convolutional neural network to classify P300 EEG signals and recognition. The experiment results show that, with the same experimental data and the number of experiments, the classification accuracy of the PCA-CNN algorithm is one of the best among algorithms SVM [21], CNN-1 [24], and BN3 [28].

For dataset I and dataset II, the algorithm produced some identical and a few different results. In both datasets, the PCA-CNN has higher accuracy rates of characters recognition than the other three algorithms (BN3 [28], CNN-1 [24], and SVM [21]) with repeat number increases, and can obtain the highest accuracy rate of character recognition in the last repeat. However, in dataset I, the comparison of the classification accuracy results of the algorithm PCA-CNN and the other three algorithms (BN3 [28], CNN-1 [24] and SVM [21]) is not as obvious as that in the dataset II. This difference may be due to the different number of subjects in dataset I and dataset II. There are only two subjects in dataset I. Thus, too small sample size may lead to insignificant differences in the comparison of classification accuracy results. In both dataset I and dataset II, the ITR value of the PCA-CNN is higher than that of the other three algorithms (BN3 [28], CNN-1 [24] and SVM [21]) in most repeats. The difference is that the comparison of the ITR value results of the algorithm PCA-CNN and the other three algorithms (BN3 [28], CNN-1 [24] and SVM [21]) is not obvious in two datasets. One possible reason is related to the difference of classification accuracy in the two datasets. According to the ITR calculation formula, its value is proportional to the classification accuracy. In dataset I, the classification accuracy rate of the PCA-CNN algorithm is not significantly higher than that of other algorithms in data II, at the maximum repeat.

The PCA algorithm was widely used in various fields of research data analysis, especially suitable for analyzing two-dimensional data matrix [38]. Researchers found that PCA could well express the basic features of the original data with less data [39]. Therefore, P300 EEG signals still retain the integrity of original EEG signals after PCA dimensionality reduction. Although the convolutional neural network can also directly extract features by itself, we find that the classification accuracy rates are higher after using PCA. As shown in Table 5, when the PCA algorithm is not added in the classification process, the accuracy of classification is lower than the accuracy of adding the PCA algorithm.The average recognition accuracy rates (subject A and subject B in the dataset I) of the PCA-CNN algorithm is 97%, while the average accuracy of recognize characters without the PCA algorithm is 94%. The former is 3% higher than the accuracy in the latter category. In Figure 6, the average accuracy rate (ten subjects in the dataset II) of recognized characters related to the CNN

algorithm which added the PCA algorithm is 90%, while the average accuracy of recognize characters without the PCA algorithm is 80%. The former is 10% higher than the accuracy in the latter category. These may be caused by the overfitting problem when the convolutional neural network processes a large amount of data, which affects the experimental performance. Adding the PCA algorithm in the convolutional neural network may solve this problem that has been proven by some literature [40,41].

**Table 5.** The comparison of PCA and NO-PCA character recognition accuracy rate in dataset I.

| Methods | Subjects | Accuracy Rate |
|---------|----------|---------------|
| **PCA** | A | **98** |
|         | B | **96** |
| **NO-PCA** | A | 96 |
|            | B | 93 |

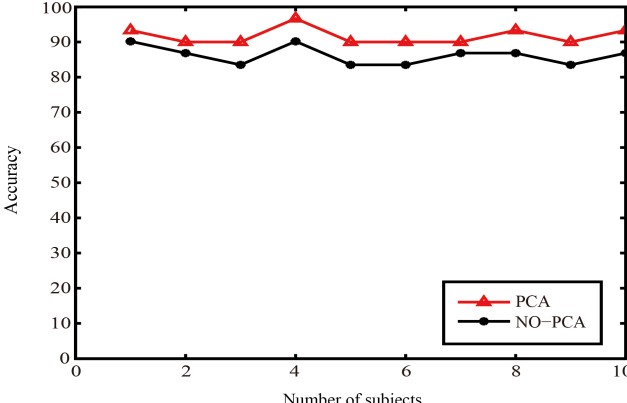

**Figure 6.** The comparison of PCA and NO-PCA characters recognition accuracy rate in dataset II.

In order to obtain a classifier with better classification ability, we improved the convolutional layer of the traditional convolutional neural network to a parallel convolution layer [42]. The parallel convolution layer adds multiple convolution kernels of different sizes to filter out the P300 EEG signal features. The number of convolution kernels determines the output of the convolution layer, so the convolution layer needs to appropriately increase the number of convolution kernels in order to more fully extract the features of signals [43]. In the previous works on the convolution neural network, such as CNN-1 [24] and BN3 [28], a single convolution kernel layer was used in the convolution part. However, when the number of signals is too large, the effect of a single convolution kernel layer in filtering features will become worse, and it is easy to ignore some features. The parallel convolution layer could increase the data capacity of the network, and may overcome the lack of features caused by improper selection of the convolution kernel size [44].

Based on the traditional convolution neural network, this paper constructed the algorithm PCA-CNN, a new algorithm for the P300 EEG signals classification. Compared with some traditional convolutional neural algorithms, the PCA-CNN algorithm has a higher classification accuracy rate for the P300 EEG signals' recognition. As shown in Tables 3 and 4, and Figure 3, the results show that the PCA-CNN algorithm has good classification performance in datasets I and II. As shown in Figures 4 and 5, the ITR of PCA-CNN is higher than the traditional SVM classification algorithm, which proves the stability of the algorithm performance. The PCA-CNN algorithm can obtain the classification accuracy rate higher than 90% on both datasets I and II; when the classification accuracy rate of P300 EEG signals is higher than 80%, the classification algorithm is effective [45]. The proposed PCA-CNN algorithm can be employed in a brain–computer interface system, and be applied for people with disability in daily lives.

## 5. Conclusions

Our work focuses on P300 EEG signals preprocessing and the convolution neural network structure designing. In the P300 EEG signals preprocessing part, the PCA is used to retain the data features of the original P300 EEG signals, which reduced the dimension of the original signals and reduced the computational cost of subsequent algorithms. In the convolution neural network structure designing part, this paper used a deep convolutional neural network to implement the classification and recognition of P300 EEG signals. The convolution neural network uses its own powerful feature extraction capabilities to construct a better classifier. The new algorithm changed the single-kernel convolutional layer in the convolution neural network to a multi-kernel convolutional layer, that is, using a multi-kernel convolution filter to extract P300 EEG signals, which improved the classification ability of the network. Compared with some traditional classification algorithms, the PCA-CNN algorithm has a higher character recognition accuracy rate. In the future, our research will consider how to improve the recognition speed of BCI system and implement an online P300 brain–computer interface system based on the deep convolution neural networks.

**Author Contributions:** Investigation, Y.X.; Methodology, X.L.; Supervision, F.L.; Validation, F.H.; Writing (original draft), X.L.; Writing (review and editing), F.W. and D.Z. All authors have read and agreed to the published version of the manuscript.

**Funding:** This research was supported by the National Natural Science Foundation of China (Grant No. 61906019), the Natural Science Foundation of Hunan Province, China (Grant No. 2019JJ50649), the Scientific Research Fund of Hunan Provincial Education Department (Grant Nos. 18C0238 and 19B004), the "Double First-class" International Cooperation and Development Scientific Research Project of Changsha University of Science and Technology (No. 2018IC25), and the Young Teacher Growth Plan Project of Changsha University of Science and Technology (No. 2019QJCZ076).

**Conflicts of Interest:** The authors declare no conflict of interest.

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
