# Peer review of "A Novel P300 Classification Algorithm Based on a Principal Component Analysis-Convolutional Neural Network"

_applsci, doi:10.3390/app10041546_

Round 1

Reviewer 1 Report

The manuscript deals with a new signal recognition algorithm for the brain-computer interface which is based on an improved convolutional neural network.

In spite of the fact that the paper is innovative and contains new results, I have some remarks:

Abstract:

It is not common to use abbreviations in the abstract, please avoid using abbreviations in the abstract.

Do not use „disabled” as an attribute of a person or people. (See page1, line 14; page1, line 25; page 11, line 232). Instead of it please write a person or people with disability.

Introduction:

It is clear and contains the relevant references.

Method:

page 5, below figure 2: „L5 and L6: Same as L3 and L4 layers.” is written. Here the predicate of the sentence is missing.

page 6, above the 13 formula: „the judgment is as follows.” is written. The sentence should begin with a capital letter.

page 9, Figure 3: Figures of Figure 3 are too small. it is hard to realize the curves and data. Please, enlarge them. The paper has no page limit.

page 10, Figure 5: The title of this figure contains only 1 bracket and it does not have an opening bracket.

Discussion:

It is written clearly, but do not write „disabled people” See my remark above in the abstract section.

Questions:

You have mentioned the data were collected in two institutes: the first one was provided by the Wadsworth Research Center NYS Department of Health, the second one was provided by the South China University of Technology. I wonder about the test people’s ability. I do not think that the BCI data are independent of their abilities.

To sum it up, the paper is written well, I suggest it for publication after minor rewriting see, my remarks above, and an English language correction is missing.

Reviewer 2 Report

(Overall) I found out there are a lot of subjective words (e.g. excellent, very appealing and effective, slightly, certain, etc.) used in the manuscript. This should be avoided or minimized.

(Fig,1) The specific number in the layer structure seems to be only applicable to dataset 1. This should be mentioned in the caption or revised.

(Discussion) Why did you have different results (accuracy and ITR) with dataset I and II? I think this should be discussed.

(Line 183) "As shown in Figure 4 and Figure 5, the PCA-CNN slightly inferior to the BN3 in the characters recognition speed, but the characters recognition speed of PCA-CNN is much higher than that of other two algorithms": The discrepancy between PCA-CNN and other two algorithms(CNN-1, SVM) seems different in Fig.4 and Fig.5. How can you say that one is a slight difference and the other is much difference (when I see the result after 15 repeats)? Are you seeing the result at other repeat numbers as well? You mentioned that PCA-CNN algorithm has advantage over other algorithms in terms of speed (due to reduction of dimension); however, the speed of recognition seems to be inferior to BN3. Can you justify this? Can you provide recognition speed data (in addition to ITR) as well?

(Line 200) "Although the convolutional neural network can also directly extract features by itself, we find that the classification accuracy rates are higher after using PCA.": Did you confirm this with dataset I as well?
